# Effects of Multi-Components on the Microwave Absorption and Dielectric Properties of Plasma-Sprayed Carbon Nanotube/Y_2_O_3_/ZrB_2_ Ceramics

**DOI:** 10.3390/nano11102640

**Published:** 2021-10-07

**Authors:** Rong Li, Yuchang Qing

**Affiliations:** 1Xi’an Research Institute of High Technology, Xi’an 710025, China; xss760829@163.com; 2State Key Laboratory of Solidification Processing, School of Materials Science and Engineering, Northwestern Polytechnical University, Xi’an 710072, China

**Keywords:** Y_2_O_3_/ZrB_2_/CNT ceramics, plasma spraying, dielectric property, microwave absorption

## Abstract

Carbon nanotube (CNT)-reinforced Y_2_O_3_/ZrB_2_ ceramics were fabricated via planetary ball milling and atmospheric-pressure plasma spraying for the first time. The phase composition, micromorphology, and electromagnetic (EM) wave absorption performance of the Y_2_O_3_/ZrB_2_/CNT hybrid was investigated from 8.2 to 12.4 GHz. Both the real and imaginary parts of the complex permittivity were enhanced as the ZrB_2_ and CNT content increased. The Y_2_O_3_/ZrB_2_/CNT hybrids corresponded to a ZrB_2_ content of 15 wt.%, and the CNT content was 2 wt.% and showed an exceptional EM wave absorption capability, with a minimum reflection loss of −25.7 dB at 1.9 mm thickness, and the effective absorption band was in a full X-band. These results indicate that an appropriate CNT or ZrB_2_ content can tune the complex permittivity and absorption performance of the Y_2_O_3_/ZrB_2_/CNT ceramics.

## 1. Introduction

Electromagnetic (EM) pollution has become a serious problem to human health and the environment due to the growth of the electronic industry and radio communication technologies [1,2,3]. To solve this problem, designing and fabricating suitable microwave absorption materials is an urgent task for researchers in the field of materials science [4,5,6,7,8]. Currently, polymer-based absorbing materials used as fillers have limited utility under special environments due to their relatively poor mechanical performance. In contrast, ceramic-based absorbers have received extensive attention due to their excellent mechanical properties [9,10,11].

Y_2_O_3_ is a sintering aid that can be added into ceramics such as SiC, SiAlON, and Al_2_O_3_; this doping promotes the solid-phase sintering of materials and can effectively improve the mechanical properties of ceramics [12,13]. In addition, zirconium diboride (ZrB_2_) has a high strength and electrical conductivity (∼6 × 10^4^ Ω^−1^·cm^−1^), high melting point (3200 °C), good thermal shock resistance, and good oxidation resistance at a high temperature. Thus, it has attracted attention as a microwave-absorption material [14].

Such excellent features can be attributed to the unique crystal structure of ZrB_2_. Typically, the graphite-like layered structure of B atoms and the electronic structure of the Zr outer layer leads to the formation of highly conductive ZrB_2_ [15]. The high conductivity implies a large value of ε″ so that the material has a good dielectric loss capability, thus causing the EM energy to be dissipated in the form of Joule heat by inducing a high current [16]. However, the strong covalent bonds of ZrB_2_ imply that hot pressing or spark plasma sintering at temperatures over 2000 °C are usually required to meet the densification of ZrB_2_-based ceramics [17]. To solve the densification problem, coatings deposited by the atmospheric-pressure plasma spraying (APS) process can effectively tune the multilayer structure and porosity and thus obtain a high bonding strength; this has become a key method for preparing ZrB_2_-based ceramic coatings.

APS outperforms traditional methods of ceramic coating, such as thermal spraying [18], laser cladding [19], micro-arc oxidation [20], and combustion synthesis [21]. It is an exceptional technology that can meet commercial requirements due to its reduced cycle times, high deposition efficiencies, low cost, and excellent microwave absorption performance. More importantly, the integration of highly conductive carbon nanotubes (CNTs) can be further optimized to control the complex permittivity and the microwave-absorption capabilities of the Y_2_O_3_/ZrB_2_ ceramic [22,23,24].

Here, Y_2_O_3_/ZrB_2_/CNT hybrids with a low CNT and ZrB_2_ content were fabricated by APS technology for the first time. The phase composition, micromorphology, and EM properties of the hybrid were investigated via X-ray diffraction (XRD), scanning electron microscopy (SEM), and a vector network analyzer (VNA), respectively. Comparative analysis of the EM properties of ceramics doped with different contents of ZrB_2_ and CNT showed that the plasma-sprayed Y_2_O_3_/ZrB_2_/CNT hybrids, corresponding to a ZrB_2_ content of 15 wt.% and the CNT content of 2 wt.%, have the best EM wave-absorption capability, with a minimum reflection loss (RL_min_) of −25.7 dB at 1.9 mm thickness, as well as an effective absorption band (EAB) in the full X-band. Therefore, Y_2_O_3_/ZrB_2_/CNT ceramics prepared via plasma-spraying could provide a theoretical basis and technical support for mass production in the field of ceramic-based high-temperature materials.

## 2. Experimental

Y_2_O_3_ (99.99%), ZrB_2_ (99.5%) and CNT were used as raw materials. The Y_2_O_3_/ZrB_2_/CNT ceramics were fabricated with different Y_2_O_3_ contents (76, 78, 81, 83, and 85 wt.%); the corresponding samples were denoted as Y76, Y78, Y81, Y83, and Y85 for simplicity. The mass percentages (wt.%) of the different Y_2_O_3_/ZrB_2_/CNT ceramics are listed in Table 1. The desired amount of Y_2_O_3_, ZrB_2_ particles, and CNT were evenly mixed using planetary ball milling. The fluidity of the powder was increased with spray drying technology, leading to spray-dried powders with a size of 30–80 μm. The Y_2_O_3_/ZrB_2_/CNT ceramic coating was prepared via the APS process, in which Ar and N_2_ were used as plasma gases, with flow rates of 50 and 10 standard liters per minute (slpm), respectively. Experimental parameters for the spray conditions were optimized for good deposition efficiency. The net arc input power and powder flow rate were 25 kW and 4 g/min, respectively; the spraying distance and coating thickness were 100 mm and 2 mm, respectively. Finally, the ceramic samples were polished to a thickness of 1.2 mm for measuring EM parameters.

The phase compositions of the plasma-sprayed samples were characterized through XRD with Cu-Kα radiation. The microstructures of the samples were observed by SEM (JEOL JSM-6360LV). The EM parameters of the Y_2_O_3_/ZrB_2_/CNT specimens with a dimension of 22.86 mm (length) × 10.16 mm (width) × 1.2 mm (thickness) were investigated in a full X-band (8.2–12.4 GHz) by VNA (Agilent technologies E8362B).

## 3. Results and Discussion

Figure 1 shows the XRD pattern of the plasma-sprayed Y_2_O_3_/ZrB_2_/CNT samples. The different Y_2_O_3_/ZrB_2_/CNT ceramics were only composed of Y_2_O_3_, ZrB_2_, and CNT crystalline phases. The peak intensity of Y_2_O_3_ obviously but gradually decreased with a decreasing Y_2_O_3_ content. The peak intensity changes in the ZrB_2_ and CNT were the same as Y_2_O_3_ when studied in the Y_2_O_3_/ZrB_2_/CNT specimens. There was no oxide peak for ZrB_2_ in the XRD spectrum, suggesting that the plasma-spray parameters were suitable, and the deposition efficiency was satisfactory.

SEM images of Y_2_O_3_/ZrB_2_/CNT ceramics are shown in Figure 2. ZrB_2_ and CNT were evenly mixed with the Y_2_O_3_ matrix. The presence of a microporous structure during the plasma-spraying process may be related to unmelted agglomerates or resolidified agglomerates. Here, the theoretical density of Y_2_O_3_/ZrB_2_/CNT ceramics could be deduced according to the density of ZrB_2_ (6.08 g·cm^3^) and Y_2_O_3_ powder (5.01 g·cm^3^). Correspondingly, the relative densities of the Y_2_O_3_/ZrB_2_/CNT hybrid decreased with the Y_2_O_3_ powder content. Furthermore, the decrease in Y_2_O_3_ content increased the porosity of the material, but the increase in CNT content better filled the pores of the Y_2_O_3_/ZrB_2_/CNT ceramics (Figure 2a,b versus Figure 2c,d).

In general, EM waves can induce two kinds of currents in the absorber: conduction current and displacement current. Within that, the conduction current comes from moving charge carriers and increased conductivity, which in turn, increases the value of ε″. The ε″ value is then related to the dielectric loss (tan δ_ε_ = ε″/ε′) capability of the absorber [25,26]. The displacement current is impacted by the dielectric polarization of local charge carriers, and the enhanced ε′ value can be attributed to the dielectric polarization and space charge polarization effects [27,28].

Figure 3 presents plots of ε′ and ε″ over 8.2 to 12.4 GHz, as the content of Y_2_O_3_ changed. In the full X-band, both ε′ and ε″ of the five samples slightly decreased as the frequency increased. In the Y85 sample, the ε′ was about 12.5, and the ε″ was only 0.8, indicating that the high content of Y_2_O_3_ and the relatively low content of ZrB_2_ lead to lower complex dielectric properties when there was no CNT. This also shows that the Y85 sample may possess a lower dielectric loss ability and poor absorbing ability.

When the ZrB_2_ content was 15 wt.%, the ε′ and ε″ values of the Y83 sample were slightly enhanced; the Y81 sample had significantly improved ε′ and ε″ values. We made a control Y85 ceramic without CNT, and we found that the ε′ and ε″ of Y85 could increase from 12.8 to 34.6 and 0.8 to 18.0, respectively. When the content of ZrB_2_ was 20 wt.%, the ε′ and ε″ of Y78 were higher than the Y83 sample. The relationship of Y76 and Y81 was the same as the result of CNT at 2 wt.%. These data indicated that Y76 had the highest values of ε′ and ε″: 41.1 and 23.4 versus the other four samples. The Y85, Y83, and Y81 samples had ε′ and ε″ values that gradually increased as the content of Y_2_O_3_ decreased. These changes were attributed to the enhancement of the CNT content, which in turn, increased the conductivity of the material and improved the dielectric properties of Y_2_O_3_/ZrB_2_ ceramics.

The ε′ and ε″ values increased significantly in Y83 and Y78 samples as the content of Y_2_O_3_ decreased at a constant CNT content. The increased ZrB_2_ content increased the complex permittivity of the material. Y81 and Y76 samples showed a similar trend. In the Y78 and Y76 samples, the ε′ and ε″ values of the materials exhibited a significant increase when the content of ZrB_2_ (20 wt.%) remained the same, due to the increase in the content of highly conductive CNTs. These results show that increasing the content of ZrB_2_ and CNT improves the ε′ and ε″ of the Y_2_O_3_/ZrB_2_/CNT ceramic. The improved conductivity of the mobile electrons in the absorber was affected by defects such as dangling bonds and vacancies, especially those caused by the interface between the ZrB_2_ and Y_2_O_3_ phases [29,30,31]. Under the effect of the EM field, free carriers in the absorber could accumulate at these interfaces, thus leading to space charge polarization and an increasing ε′. Therefore, the enhanced ε′ value was related to an increased ZrB_2_ content and the number of ZrB_2_/Y_2_O_3_ interfaces. Furthermore, a higher ZrB_2_ and CNT content would also lead to more permeable electrical pathways, thus improving the ε″ value. Moreover, when the content of CNT was equal, the enhanced ZrB_2_ content increased the value of ε′ and ε″. The increase in CNT content enhanced the ε′ and ε″ values of the Y_2_O_3_/ZrB_2_/CNT ceramic when the content of ZrB_2_ was equal. Therefore, we concluded that the presence of ZrB_2_ or CNT enacted a significant effect on the dielectric properties of Y_2_O_3_/ZrB_2_/CNT ceramics.

The EM wave absorption performance of the plasma-sprayed Y_2_O_3_/ZrB_2_/CNT ceramic coatings from 8.2 to 12.4 GHz are shown in Figure 4a–e. In Figure 4a, the minimum reflection loss (RL_min_) of the Y85 sample was only −3.8 dB at 9.4 GHz, with a 2.2 mm thickness. The Y83 sample (Figure 4b) had a RL_min_ value of −25.7 dB at 12.3 GHz, with a 1.9 mm thickness. The improved microwave absorption ability may have been caused by the enlarged ε′ and ε″ values from CNTs, forming a dense conductive percolation network. This network increased the conductivity loss. Furthermore, the effective absorption band (the RL_min_ was lower than −10 dB) of Y83 ceramics covered the full X-band when the absorber’s thickness was 2.0 mm. However, the Y81 ceramic showed an RL_min_ value of −13.0 dB at 1.3 mm thickness from 8.2 to 12.4 GHz (Figure 4c). The decreasing RL_min_ value may be attributed to the highest conductivity based on excessive CNT content. Excessive CNTs cause more EM waves to be reflected from the emitting surface; thus, the waves cannot be absorbed and consumed, i.e., impedance mismatching.

When the ZrB_2_ content increased to 20 wt.%, the RL_min_ values of Y78 and Y76 were −22.0 dB at 12.2 GHz with a 1.9 mm thickness and −9.8 dB at 12.4 GHz with a 1.2 mm thickness, respectively (Figure 4d,e). Such results further prove that an appropriate CNT content is critical to an optimized absorbing performance in Y_2_O_3_/ZrB_2_/CNT ceramics. Therefore, the excellent RL value of the Y_2_O_3_/ZrB_2_/CNT material corresponded to the ZrB_2_ content (15 wt.%) and the CNT content of 2 wt.%. The loss tangent (tan δ) of the prepared composite samples, which is representative of the power loss in the material with respect to the stored reactive power, was estimated as the ratio between the imaginary and real part of the complex effective permittivity [32,33]. The computed values of tan δ are reported in Figure 4f. We notice that tan δ, which increased with the contents of ZrB_2_ and CNT in the composite, was always greater than Y85 in the whole frequency range for samples Y76, Y78, Y81, and Y83. As we know, the highest tan δ_ε_ values for the sample represent the impressive storage and loss capabilities for electrical energy. The dielectric loss behavior is mainly related to conductivity loss and polarization loss. Among that, ionic polarization and electronic polarization always work in the frequency of 10^3^–10^6^ GHz, and dipole orientation polarization from the frustrated reorientation of dipoles prefers frequency dispersion [34,35]. Thus, the dielectric loss should be determined by the interfacial polarization between Y_2_O_3_, ZrB_2_, and CNT. Furthermore, the highest dielectric loss capability prompts it to consume electromagnetic waves and obtain an excellent reflection loss.

To better analyze this phenomenon of different microwave absorbers, two essential factors, the EM attenuation constant (α) and impedance matching ($\left|Z_{in}/Z_{0}\right|$), need to be considered [36,37,38]. The maximum value of α suggests that more EM waves can enter the material and transform EM energy into thermal energy, as shown in Equation (1) [39]:(1)$$\alpha =\frac{\sqrt{2}}{c}\pi f\times \sqrt{(\mu^{\prime \prime }\varepsilon^{\prime \prime }-\mu^{\prime }\varepsilon^{\prime })+\sqrt{(\mu^{\prime \prime }\varepsilon^{\prime \prime }-\mu^{\prime }\varepsilon^{\prime }{)}^{2}+(\mu^{\prime \prime }\varepsilon^{\prime \prime }+\mu^{\prime }\varepsilon^{\prime }{)}^{2}}}$$

Figure 5a presents the α value of the five samples in the X-band. The Y85 sample showed an average α value of about 19.0, but the α value of doped-CNT samples had a major effect relative to the lowest α value seen in the undoped-CNT Y85 ceramic. These data further prove that the addition of CNT can improve EM attenuation, leading to good microwave absorption. The average α values of Y76 and Y81 were 319.8 and 267.1, respectively. The average α values of Y78 and Y83 were 188.8 and 176.4, respectively. The Y76 and Y81 values showed a larger α value versus Y78 and Y83, suggesting an exceptional EM wave attenuation ability. However, Y78 and Y83 ceramics had higher reflection loss values than Y76 and Y81: the attenuation constant only indicates that the material has a strong EM attenuation ability. No other factor could be considered, i.e., impedance matching. Figure 5b–f shows the impedance matching of five samples simulated at different absorbing thicknesses. Values of $\left|Z_{in}/Z_{0}\right|$ of Y85, Y81, and Y76 ceramics were far from 1, but Y83 and Y78 approached 1 as the absorbing thickness changed. These observations suggest that Y83 and Y78 samples absorb more EM waves. Therefore, good microwave absorption is related to both EM attenuation loss and impedance matching.

## 4. Conclusions

Y_2_O_3_, ZrB_2_, and CNT powders were uniformly mixed via a planetary ball mill, followed by spray drying. The Y_2_O_3_/ZrB_2_/CNT ceramics were fabricated by APS technology. The ε′ and ε″ values of Y_2_O_3_/ZrB_2_/CNT ceramics increased when the ZrB_2_ or CNT contents increased, because Y_2_O_3_/ZrB_2_/CNT ceramics have dense conductive percolation pathways. These pathways can respond to the presence of conductance losses and thus improve the microwave absorption ability. The RL_min_ value of the Y83 sample was −25.7 dB at 12.3 GHz with a 1.9 mm thickness. One could control the EM properties of Y_2_O_3_/ZrB_2_/CNT ceramics by tuning the mass percentage of Y_2_O_3_, ZrB_2_, and CNT. This could lead to a high attenuation capability and good impedance matching. This unique Y_2_O_3_/ZrB_2_/CNT hybrid had excellent EM wave absorption properties and is a good candidate for designing ceramic-based high-temperature absorbers.

## Figures and Tables

**Figure 1 nanomaterials-11-02640-f001:**
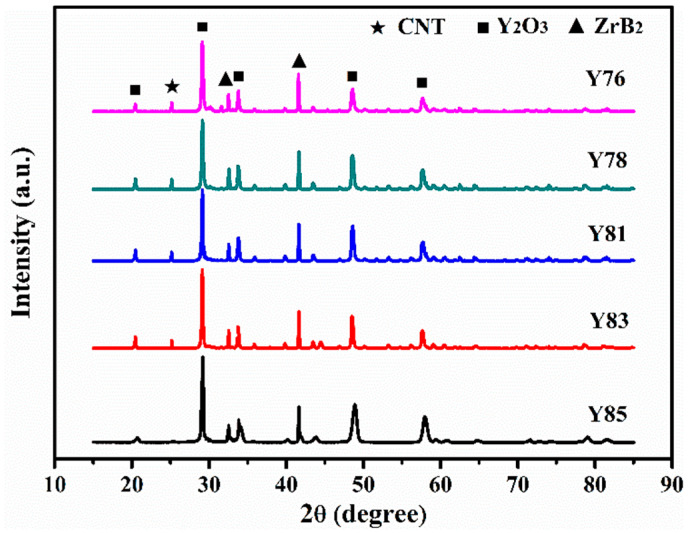
XRD spectra of Y_2_O_3_/ZrB_2_/CNT specimens with different Y_2_O_3_ contents.

**Figure 2 nanomaterials-11-02640-f002:**
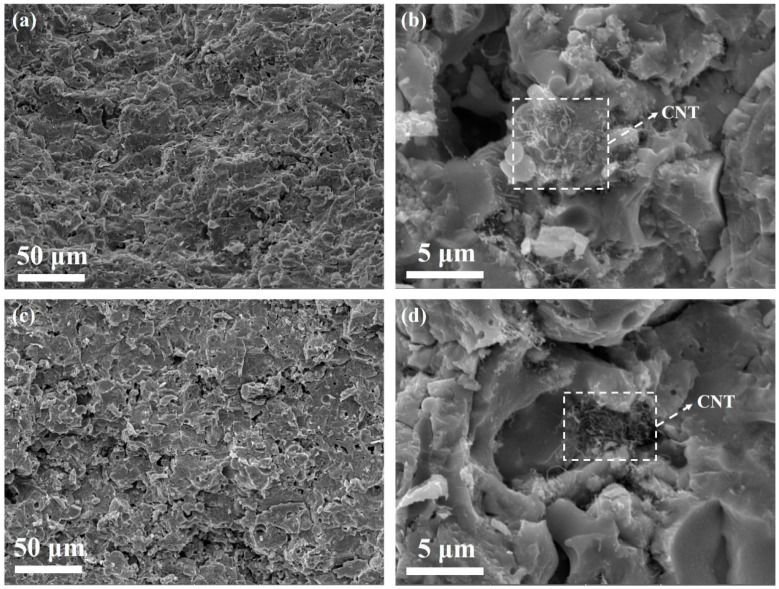
SEM images of Y_2_O_3_/ZrB_2_/CNT ceramics with (**a**,**b**) 83% Y_2_O_3_ particles (Y83 sample) and (**c**,**d**) 81% Y_2_O_3_ particles (Y81 sample).

**Figure 3 nanomaterials-11-02640-f003:**
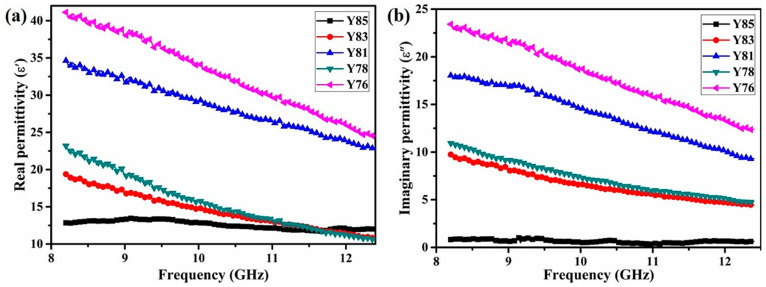
The (**a**) ε′ and (**b**) ε″ plots of the plasma-sprayed Y_2_O_3_/ZrB_2_/CNT hybrids in the full X-band.

**Figure 4 nanomaterials-11-02640-f004:**
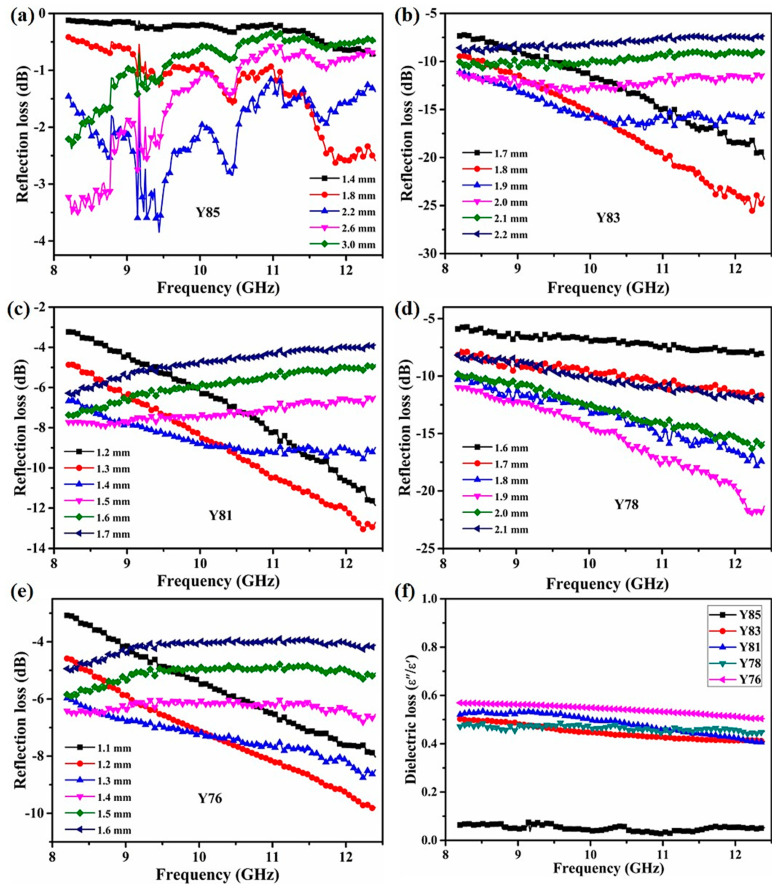
(**a**–**e**) Reflection loss with different absorber thicknesses and (**f**) dielectric loss curves of the plasma-sprayed Y_2_O_3_/ZrB_2_/CNT hybrids.

**Figure 5 nanomaterials-11-02640-f005:**
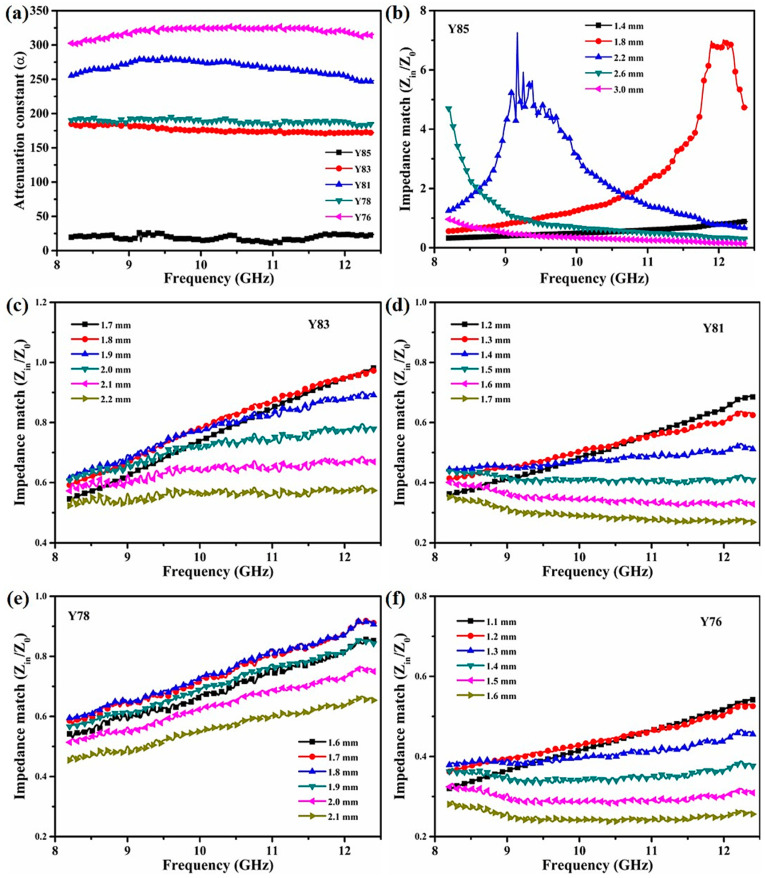
Frequency dependence of (**a**) attenuation constant and (**b**–**f**) impedance matching of Y_2_O_3_/ZrB_2_/CNT ceramics with the different absorber thicknesses.

**Table 1 nanomaterials-11-02640-t001:** The mass percentage (wt.%) for different Y_2_O_3_/ZrB_2_/CNT ceramics.

Sample	Y_2_O_3_ (wt. %)	ZrB_2_ (wt. %)	CNT (wt. %)
1	85	15	-
2	83	15	2
3	81	15	4
4	78	20	2
5	76	20	4

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
