# Peer review of "Effects of Multi-Components on the Microwave Absorption and Dielectric Properties of Plasma-Sprayed Carbon Nanotube/Y2O3/ZrB2 Ceramics"

_nanomaterials, 2021, doi:10.3390/nano11102640_

Round 1

Reviewer 1 Report

Peer Review of manuscript nanomaterials-1398773, Rong and YuChang, “Effects of multi-components on the microwave absorption and dielectric properties of plasma-sprayed carbon nanotube/Y2O3/ZrB2 ceramics.”

The authors describe a composite ceramic with application to microwave absorption, which is measured in the 8-12 GHz X-band.  An optimization study is performed based on varying the fractions for ZrB2 and carbon in the Y2O3 binder.  The results are potentially useful.  We have several suggestions for improving the manuscript.

The motivation statement that ‘EM pollution has become a serious problem to human health’ needs to be supported by a significant reference, such as a review article or editorial from a major journal like Science or Nature.  If that can’t be done, then the problem is probably not “serious”, and might be just “potential”.  

Inconsistencies between sections.  For example, the ceramic samples in experimental details section are stated to have been polished to 1.2 mm.  However, in results and discussion section, the sample thicknesses range from 3 mm to 1.1 mm (figure 4).

Thickness is a parameter that affects the measured absorption.  Since thickness is varied, it weakens, or at least confuses, their conclusions about the dependence on ZrBr2 and C composition.  Absorption coefficient, on the other hand, is independent of thickness.  Absorption coefficient would seem to be the one figure of merit that is directly related to the application.  A discussion of the dependence of its band-averaged value without all the other confusing details would simplify and clarify the result.  A subset of the spectra could be presented as examples.  They don’t all need to be shown.

If loss tangent is also important, that can be compared, too, but explain why it is independently interesting.

Is reflectivity important to the application?  Explain why.  If so that can be compared, too.

Is impedance match the important quantity?  Explain why.  Just because a quantity can be determined from a result, doesn’t make it an important figure of merit for a specific application. 

As far as we can tell from the manuscript, only absorption coefficient and reflectivity are important figures of merit for the stated application.

Real and imaginary parts of the permittivity are uninteresting by themselves, from the point of view of the application.  Only example spectra need be presented.  Their chief value is that from them can be calculated absorption coefficient, loss tangent, and reflectivity.

Equation 1 gives the absorption coefficient as a function of complext permittivity and permeability.  Is permeability different from unity for this material at X-band?  No data for it is presented.  Therefore, we have to assume that Equation 1 is too general, and a simpler formula can be given.  And the reference to it should be a book, not an obscure research article.  For instance:  “Absorption coefficient determines the exponential decay of the intensity in the medium and equals twice the imaginary part of the wavevector.  That is (2 w/c) times the extinction coefficient kappa, where kappa^2 = (1/2)(-e’+Sqrt[e’^2+e”^2]).  The w = angular frequency and e is permittivity.  A suitable reference would be (e.g.) LD Landau, EM Lifshitz, and LP Pitaevskii, Electrodynamics of Continuous Media, 2nd Ed (Elsevier, Butterworth, Heinemann, 1984) section 83.

There are some additional small changes that would add clarity.

Line 55: instead of ‘in the first time’, use ‘for the first time’

Line 57: expanded form definition for XRD, SEM and VNA

Line 57-62: Introduction states that that the best EM wave-absorption and minimum reflection loss of 25.7 dB is reported at 1.9 mm thickness. This appears to be theoretically deduced result using equation 1. Because in experimental section, line 78 states that ceramic samples were polished to a thickness of 1.2 mm. If all samples were polished to the thickness of 1.2 mm, it is recommended that it must be stated unequivocally in experimental section.

Line 81-82: there are multiple samples, change ‘plasma-sprayed sample was characterized’, to ‘plasma-sprayed samples were characterized’

Line 82-83: because multiple samples were characterized (at least two, that are shown in figure 2), statement that represents plurality of the samples may be more appropriate and clear to the readers.

Line 81-86: acronyms XRD, SEM and VNA defined in line 57 may be used here

Line 112: Please change ‘increases conductivity’, to ‘increased conductivity’

Figure 5(a).  The vertical axis is labeled “Attenuation constant”, but what is meant is “Absorption coefficient”.

Reviewer 2 Report

Ref.comments to the paper titled as “Effects of multi-components on the microwave absorption and dielectric properties of plasma-sprayed carbon nanotube/Y2O3/ZrB2 ceramics” written by the authors Li Rong1 and Qing Yuchang.

It is known that study of the different ceramics, which can replace the crystals or semiconductors are really perspective at the present time. MgF2, ZnS, other types of these materials are investigated intensively. From this point of view, the paper devoted to consider the properties of the Zr-based ceramics modified with the CNTs is accrual and modern.

For the first, it is remarked that the authors have made nice literature search, analyzing 34 references. Moreover, the papers written by the last 3 years are analyzed as well. Good! This indicates the knowledge of the problem, its useful application and finding ways to solve it.

The preparation process is written clear; the methods and approaches are adequate. The results and discussion parts are coincided with our physical knowledge.

The real and imaginary parts of the dielectric constant have been studied for the modified ceramics; reflection losses with different absorber thicknesses have been established; frequency dependence of attenuation constant and impedance matching of Y2O3/ZrB2/CNT ceramics with the different absorber thicknesses have been estimated. But, I would like the authors what is the value of the refractive index of your modified ceramics, taken into account that the refractive index of the CNTs is close to 1,1? How this value will coincide with the Fresnel losses?

So, the paper is interesting for the specific area for the researchers. A little remark: I can recommend to the author to answer the questions mentioned above.  Thus, the paper can be published after minor corrections.
